# Assessing the level of the material deprivation of European Union countries

**Aleksandra Łuczak**[1], **Sławomir Kalinowski**[2]*

**1** Department of Finance and Accounting, Poznan University of Life Sciences, Poznań, Poland,
**2** Department of Rural Economics, Institute of Rural and Agricultural Development, Polish Academy of Sciences, Warsaw, Poland

* Skalinowski@irwirpan.waw.pl

## Abstract

The purpose of this paper is to assess the level of material deprivation in European Union countries in 2016 from both a local and a global perspective. The Technique for Order Preference by Similarity to an Ideal Solution (TOPSIS) was used in the study. Based on research, five main types of the level of the material deprivation of European Union countries were identified. Research findings suggest that the population of old EU countries is less severely affected by material deprivation than people living in new member states. Also, the level of global material deprivation was assessed. The study was based on 2016 statistical data delivered by Eurostat.

**Citation:** Łuczak A, Kalinowski S (2020) Assessing the level of the material deprivation of European Union countries. PLoS ONE 15(9): e0238376. https://doi.org/10.1371/journal.pone.0238376

**Data Availability Statement:** All relevant data are within the manuscript and its Supporting Information files.

**Funding:** Source of funding – article fee: Poznań University of Life Sciences (Uniwersytet Przyrodniczy w Poznaniu), proofreader fee:

## Introduction

Combating poverty is one of the priorities set out in the Europe 2020 strategy for smart, sustainable and inclusive growth. Measures to alleviate poverty are a priority just as much as actions focused on supporting sustainable economic growth and employment, as specified in the Lisbon strategy. It was assumed that by 2020 the number of people at risk of poverty must be reduced by a total of 20 million [1] from the current level of over 33.6 million (including 27.6 rural residents). Note also that currently 48.4 million people continue to be affected either by poverty or by social exclusion (including 35.5 million rural residents). It would be difficult to consider a perfect world like More's Utopia, where well-being is an inherent part of the society. Today, social inequalities are common. It is therefore up to governments to create conditions where all citizens have their basic needs addressed [2].

Many questions on how to alleviate poverty and material deprivation remain unanswered. Meanwhile, new questions arise on how to help excluded communities, narrow the gap between countries, determine the scope and forms of aid and establish national and international support schemes to make aid a structural instrument. Although based on altruistic motives, support measures taken at national level large correlated with egoistic concerns. Indeed, aid can drive long-term reputational and economic benefits. Providing people with support and development opportunities could counteract the risk of social tensions in various EU countries. Economic development in the EU drives improvements in the standard of living while offering benefits to community members. It is a specific investment in the social well-

Institute Of Rural and Agriculture Development of the Polish Academy of Sciences (Instytut Rozwoju Wsi i Rolnictwa Polskiej Akademii Nauk). The funders had no role in study design, data collection and analysis, decision to publish, or preparation of the manuscript. The authors received no specific funding for this work.

**Competing interests:** NO authors have competing interests.

being of EU member states. If even only one member experiences an adverse economic and social economic situation, this could become a major problem for the entire EU. For the reasons set out above, it seems essential to empower the international institutions so they can effectively enforce social policy standards across the EU. Conventions and directives should make it possible to raise social standards and, as a consequence, promote the elimination of material deprivation and of large differences in the extent to which individual needs are addressed. It also seems necessary to prevent unfair practices related to the underestimation of labour costs.

Narrowing the development gap, preventing and eliminating poverty and eradicating the sources of social instability are imperatives for today's public policy [3]. The growing importance of social policies results from society becoming increasingly polarised (including the differences in the extent to which individual needs are met), both at EU and national level.

The purpose of this paper is to assess the level of material deprivation in European Union countries in 2016 from both a local and global perspective. An effort was also made to seek the relationship between the assessment of the level of material deprivation in EU countries and the at-risk-of-poverty rate. Identifying the major poverty factors could provide a basis for designing tools that reduce the risk of poverty and social exclusion in various EU countries. In this context, note that the social integration process in the EU is based on the open method of coordination (OMC), which provides member states with considerable discretion in choosing the measures and priorities when carrying out their tasks. Hence it is important to identify the levels of material deprivation in different countries of the European Union as one of the methods for diagnosing their social situation.

The authors' contribution includes, firstly, a proposal for quantitative approaches based on the Technique for Order Preference by Similarity to an Ideal Solution (TOPSIS) to assess the material deprivation level in EU countries. Secondly, the authors have improved the quality of assessment of the material deprivation level in EU countries and expanded possibilities of research from two perspectives (local and global). The local approach presents the relative situation of countries in the EU. The global approach shows the absolute situation of EU countries, i.e. in a broader (global) context. The results of the study are important for decision-makers and politicians in EU who participate in the process of creating documents (polices, financial plans, strategies).

## Literature review

As emphasised by researchers, poverty is a multidimensional issue [4–11]. Identifying it requires the definition of a series of social, economic and political factors. The definition of poverty varies as a function of the social, cultural or historical context [12–17]. Two rival concepts emerge in the relevant research. One is based on material resources whereas the other relates to the actual outcomes of an individual's actions. However, in fact it is difficult to separate the two concepts, and the above approaches are considered complementary. This makes it possible to provide a broad picture of how a population lives [18]. In a more general sense, poverty may be regarded as critical circumstances that result from the failure to address an individual's needs at a socially accepted level. Such a broad approach is the opposite of poverty defined narrowly and exclusively as the unavailability of financial resources to meet one's needs. When extended to include factors other than income, the definition becomes richer and makes it possible to assess poverty in the EU on a more comprehensive basis.

Three indicators are used to assess poverty in the European Union: the relative poverty indicator (60% of national median equivalent income), the proportion of people living in households with very low work intensity (which means people living in households where

working-age members worked less than 20% of their total work potential during the previous year), and the indicator of severe material deprivation [19]. The latter presents the percentage of households which, for objective reasons, are unable to address at least one of the following nine needs: afford a one-week holiday away from home for all household members once a year; eat meat or fish (or a vegetarian equivalent) every second day; heat the apartment as needed; meet unexpected expenditure in the amount equal to the monthly threshold of relative poverty (as defined for the country concerned for the year preceding the survey); pay housing-related expenses, instalments and loans on time; own a colour TV; own a car; own a washing machine; and own a phone.

The measurement of material deprivation seems to be extremely important in the analysis of absolute poverty. This is because it does not focus on poverty symptoms in relation to other households [20–24], and therefore facilitates greater comparability of living conditions across EU countries [25]. This indicator comprises two dimensions: the economic strain, which includes the first five symptoms, and the durables, which include the remaining four symptoms. The rationale behind choosing these needs is called into question by some researchers, including Guio and Marlier [26], who suggest they should be modified to a certain extent by replacing the need to own a colour TV, a car or a washing machine by seven other needs, including: replace worn-out clothes with new ones; have two pairs of properly fitting shoes; spend a small amount of money on oneself each week; have regular leisure activities; get together with friends or family for a drink/meal at least once a month; have an internet connection; replace worn-out furniture.

The material deprivation indicator is more sensitive to differences in standards of living between the countries than the at-risk-of-economic-poverty rate, as the latter is based on relative values. In the material deprivation indicator, short-term deficiencies or insufficient income may be compensated by savings or available loans or by selling assets; a person with insufficient incomes does not necessarily experience deprivation (in this context, it is worthwhile comparing that situation with the ratchet effect, i.e. an attempt to maintain the existing standards of living despite losing some income, hoping that the situation is only transitory). This indicator also takes account of incomes in kind, free or subsidised goods and self-supply. Indeed, the consumer may actually be in a better situation than their income suggests. Or it may be the opposite. Real purchasing power may be lower because of one's family or personal situation, e.g. disability, temporary illnesses, dysfunctions etc. The above implies that research should take account of incomes equated with consumption (with ongoing consumption expenditure) in order to address the benefits derived from the household's durables (note, however, that the purchasing power of money in different countries also plays an important role). This concept is also supported by Friedman [27]. At the same time, the analyses of material deprivation should consider Mack and Lansley's [28] concept, who paid attention to a forced deficiency of goods. This approach makes it possible to exclude people who intentionally fail to address a certain group of needs from the survey [26, 29, 30].

Research on material poverty is quite frequent [21, 23, 31–38]. However, the analyses continue to focus mostly on the income-related dimension [39] while disregarding the ownership of goods; this results from data accessibility and analytic capabilities. Instead, these dimensions should be considered complementary, having in mind that material deprivation provides additional information that is ignored in the analysis of disposable income. An interesting piece of research information can be derived from the relationship between material deprivation and the at-risk-of-poverty rate represented by 60% of the national median income. The relationship makes it possible to identify two specific groups of people: low-income earners who meet their material needs and those who do not address their needs despite having relatively high incomes. These circumstances require detailed analysis in order to establish the needs that are

the least addressed and, on the other hand, to reveal the reasons why people decide not to pursue their needs. However, the concentration of material and financial issues is the greatest problem for national social policies.

The relevant literature notes that two approaches are used to explain material deprivation at a micro and macro level. At a micro level, individual deprivation is analysed and an investigation is carried out into the mechanisms that affect deprivation, setting aside the pool of national particularities. The micro dimension represents a broad approach to material deprivation, as it takes account of an extensive set of socioeconomic determinants, i.e. age [40, 41], gender [42–44], education [31, 45–47], position in the labour market [46, 48], household size [31] and incomes [46, 49–51]. Conversely, the macro dimension is based on national particularities, including: cultural factors, the extent and nature of social assistance [52], the unemployment rate and structure [41], labour-market flexibility, workforce mobility, GDP and the degree of social inequality [53–56]. Each of these dimensions affects both the perceived and the objective level of deprivation. Later on, macro-level factors will be used in this paper.

## Methods

The positional TOPSIS (Technique for Order Preference by Similarity to an Ideal Solution) approach was proposed in order to assess the level of material deprivation in European Union countries (selected non-EU but closely related countries were also taken into account). The classical version of the TOPSIS method was developed by Hwang and Yoon [57] and is one of the best-known techniques for solving multi-criteria decision-making problems with a finite number of alternatives. TOPSIS is very useful in constructing the ranking of objects described by many variables. It is based on the distances of objects from the ideal solution and the anti-ideal solution. Distances are the basis for constructing a synthetic measure. In the assessment of the material deprivation of EU countries, variables characterised by strong asymmetry and atypical values (outliers) may occur. In such cases the classical TOPSIS method may be unreliable and contribute to problems connected with the complete and accurate identification of types of material deprivation. In studies concerning material deprivation the focus should thus be on robust methods that limit the impact of strong asymmetry and outliers, particularly those using the Weber spatial median [58]. It should be added that TOPSIS and its modification and various versions have been widely used in many issues [59–61] i.e. business, management, health, safety, environment and many others.

The proposed procedure of construction of the synthetic measure based on the positional TOPSIS method includes five basic stages (Fig 1).

The first stage includes the selection of variables describing units, as well as determination of the direction of their preferences in relation to the main criterion. The selection of variables is based on substantive and statistical analysis.

The second stage consists in determining and normalising the values of variables for objects (e.g. countries of the European Union). The selected variables are classified as stimulants, destimulants and nominants. They have a stimulating or destimulating effect on the phenomenon. Variables that have stimulating effect contribute to increasing the level of the phenomenon and are called stimulants, while variables with a destimulating effect decrease it and are called destimulants. Nominants are the type of variables that are destimulants in one range of a variable and stimulants in another. Desirable (optimal) values should be defined for the nominants. A normalisation procedure is carried out in order to compare the value of the variables. Its objectives include [62]:

- enabling the comparability of variables expressed in different units (the additivity requirement),

| Stage 1 | • selecting variables on the complex phenomenon (i.e. the material deprivation of European Union countries) |
| Stage 2 | • determining and normalising the values of variables for objects (countries of the European Union) |
| Stage 3 | • calculating the distance of each object (countries of the European Union) from positive and negative ideal solutions |
| Stage 4 | • calculating values of the synthetic measure |
| Stage 5 | • linearly ordering and identifying the types |

**Fig 1. Procedure of construction of the synthetic measure.** Source: own elaboration.

- unifying the nature of variables by converting those with an inhibiting or nominant effect into variables with a stimulating effect (the unified preference requirement),

- removing non-positive values of variables from the calculations (the positive value requirement),

- replacing different ranges of variability of variables with a constant range for all variables under consideration (the requirement of constant range or constant extreme values).

For the normalisation the zero unitarisation procedure was applied based on the following formulae [63]:

for stimulants

$$z_{ik} = \frac{x_{ik} - \min\{x_{ik}\}}{\max\{x_{ik}\} - \min\{x_{ik}\}}, \tag{1}$$

for destimulants

$$z_{ik} = \frac{\max\{x_{ik}\} - x_{ik}}{\max\{x_{ik}\} - \min\{x_{ik}\}} \tag{2}$$

for nominants

$$z_{ik} = \frac{x_{ik} - \min\{x_{ik}\}}{\text{nom}\{x_{ik}\} - \min\{x_{ik}\}}, \quad x_{ik} \leq \text{nom}\{x_{ik}\}, \tag{3}$$

$$z_{ik} = \frac{\max\{x_{ik}\} - x_{ik}}{\max\{x_{ik}\} - \text{nom}\{x_{ik}\}}, \quad x_{ik} > \text{nom}\{x_{ik}\}. \tag{4}$$

with $\max\{x_{ik}\}$: maximum value of the $k^{\text{th}}$ variable; $\min\{x_{ik}\}$: minimum value of the $k^{\text{th}}$ variable; $\text{nom}\{x_{ik}\}$: nominal (optimal) value of the $k^{\text{th}}$ variable.

Instead of maximum and minimum values of variables, model values may also be used. For instance, if the assessment is made for countries within the European Union, the model objects could be the maximum and minimum values recorded in other (non-EU) countries or

absolute pattern and anti-pattern values of variables. The approach based on maximum and minimum values recorded across the units covered by the study makes it possible to determine the local (relative) situation (i.e. local level of the material deprivation). In turn, the approach based on model values shows the situation of units in a broader context–global (absolute) situation (i.e. the global level of the material deprivation).

The set of variables describing the material deprivation of European Union countries sometimes includes the variables with strong asymmetry and outliers. These observations may considerably affect the outcomes of the linear ordering of objects by the TOPSIS method. In such cases, the assumption that the maximum and minimum values of the variables (as used in the TOPSIS method) correspond, respectively, to the positive ideal solution and the negative ideal solution leads to excessive remoteness from typical values of the variables considered, and consequently narrows the range of variation of the development indicator (synthetic measure), making it difficult to correctly identify the development types of the objects examined on its basis [64]. It is important to identify the outliers and use the appropriate method. Outliers can be identified based on various statistical methods [65]. In these methods for linear ordering, ideal solutions and negative ideal solutions are set separately for each variable. The method for the identification of outliers proposed in this paper therefore relies on a single-dimensional approach: the quartile criterion. The values of a single variable are found to be outliers if located outside the following interval [66]:

$$\langle Q_{1k} - 1,5 \cdot IQR_k, Q_{3k} + 1,5 \cdot IQR_k \rangle \tag{5}$$

where $Q_{1k}$, $Q_{3k}$ –respectively the first (lower) and the third (upper) quartiles of the $k^{\text{th}}$ variable, $IQR_k$ –are the interquartile range of the $k^{\text{th}}$ variable.

The problem may be solved by an approach with the application of winsorised data for the values of each variable that are not within the above interval. The zero unitarisation procedure is calculated for winsorised data. Winsorisation is a process of replacing a specified number of outliers of variables with a constant (smaller or bigger) value.

The zero unitarisation transformation results in converting variables with an inhibiting or nominal effect into variables with a stimulating effect while also enabling the comparability of their values. In third stage the positive ideal solution (PIS) and negative ideal solution (NIS) were calculated as [62]:

$$\text{PIS} \qquad A^+ = \left( \max_i(z_{i1}), \max_i(z_{i2}), ..., \max_i(z_{iK}) \right) = (z_1^+, z_2^+, ..., z_K^+), \tag{6}$$

$$\text{NIS} \qquad A^- = \left( \min_i(z_{i1}), \min_i(z_{i2}), ..., \min_i(z_{iK}) \right) = (z_1^-, z_2^-, ..., z_K^-). \tag{7}$$

The positive ideal solution (the ideal solution, the pattern or the ideal object) are the best values of variables. The PIS includes the maximum (or ideal) values of each variables, whereas, the negative ideal solution (the anti-pattern or the anti-ideal object) is the variables' worst values. The NIS contains the minimum (nadir) values of each variable, which are stimulant or are transformed into stimulant.

Next, Euclidean distances for each unit from the PIS ($A^+$) and the NIS ($A^-$) were calculated:

$$d_i^+ = \sqrt{\sum_{k=1}^{K} (z_{ik} - z_k^+)^2} \qquad d_i^- = \sqrt{\sum_{k=1}^{K} (z_{ik} - z_k^-)^2} \tag{8}$$

Then, in the fourth stage, the values of synthetic measure are calculated [56]:

$$S_i = \frac{d_i^-}{d_i^- + d_i^+}, (i = 1, \ldots, N), \qquad 0 \le S_i \le 1. \tag{9}$$

The smaller the distance from the positive ideal solution, and thus the further from the negative ideal solution, the closer to 1 is the value of the synthetic measure.

In the fifth stage, values of the synthetic measure are used in rank-ordering of objects and are the base for identification of their typological classes. The types may be identified based either on a formal or a substantive method [67]. Formal identification involves naming the types, whereas substantive identification boils down to describing the types with selected descriptive statistics of the variables considered (usually, intra-class means). The classes may be described with endogenous variables, i.e. those involved in the creation of the synthetic measure, and with exogenous variables, which, although substantively related to the phenomenon considered, were not used as a basis for the ranking.

Formal identification of classes for the entire range of variation of a synthetic measure may be performed using statistical methods or in an arbitrary manner, assuming e.g. numerical ranges of values for the synthetic measure. The determined values of the synthetic measure are linearly ordered and become the basis for grouping the countries into typological classes by material deprivation level. The entire range of the synthetic measure was arbitrarily divided into classes. This study assumes that the values of indicator $S_i$ fall within the following numerical intervals: $\langle 0.80, 1.00 \rangle$–very high level, $\langle 0.60, 0.80 \rangle$–high level, $\langle 0.40, 0.60 \rangle$–medium level, $\langle 0.20, 0.40 \rangle$–low level, $\langle 0.00, 0.20 \rangle$–very low level.

## Results of research

The study concerning the level of material deprivation in European countries ($N = 33$) is based on statistical data for 2016 from Eurostat [68]. The study focuses on European Union members and collaborating countries which either are closely related to the EU or aspire to European integration.

In the first stage, nine variables were based on a substantive and statistical analysis. The research procedure was based on the percentage of household members who state they are unable to meet the following needs due to financial constraints:

1. afford a one-week holiday away from home for all household members once a year ($x_1$),

2. eat meat or fish (or a vegetarian equivalent) every second day ($x_2$),

3. heat the apartment as needed ($x_3$),

4. meet unexpected expenditure to the amount equal to the monthly threshold of relative poverty (as defined for the country concerned for the year preceding the survey) ($x_4$),

5. pay housing-related expenses, instalments and loans on time ($x_5$),

6. own a colour TV ($x_6$),

7. own a car ($x_7$),

8. own a washing machine ($x_8$),

9. own a (fixed or mobile) phone ($x_9$).

Table 1 presents descriptive statistics of the material deprivation variables of European Union countries in 2016 [68]. All variables demonstrated positive skewness, with strong or

**Table 1. Descriptive statistics of the material deprivation variables of European Union countries in 2016.**

| Specification | variables | | | | | | | | |
|---|---|---|---|---|---|---|---|---|---|
| | $x_1$ | $x_2$ | $x_3$ | $x_4$ | $x_5$ | $x_6$ | $x_7$ | $x_8$ | $x_9$ |
| | initial data | | | | | | | | |
| Mean | 34.25 | 9.75 | 9.86 | 37.86 | 14.47 | 0.44 | 9.25 | 1.38 | 0.51 |
| Median | 36.70 | 6.10 | 5.80 | 37.90 | 10.60 | 0.30 | 6.90 | 0.60 | 0.20 |
| Max | 66.60 | 41.80 | 39.20 | 60.00 | 47.90 | 1.40 | 32.90 | 9.80 | 3.60 |
| Min | 6.00 | 1.30 | 0.60 | 18.10 | 4.20 | 0.00 | 2.00 | 0.00 | 0.00 |
| St. Dev. | 0.53 | 0.96 | 0.99 | 0.35 | 0.77 | 0.75 | 0.76 | 1.59 | 1.51 |
| Skewness | 0.14 | 1.82 | 1.47 | 0.20 | 1.64 | 1.32 | 1.60 | 3.05 | 2.87 |
| Ex. kurtosis | -1.14 | 3.57 | 1.40 | -1.29 | 1.96 | 1.29 | 2.63 | 9.37 | 8.94 |
| | winsorised data | | | | | | | | |
| Mean | 34.25 | 8.83 | 8.57 | 37.86 | 12.58 | 0.41 | 8.32 | 0.84 | 0.36 |
| Median | 36.70 | 6.10 | 5.80 | 37.90 | 10.60 | 0.30 | 6.90 | 0.60 | 0.20 |
| Max | 66.60 | 23.03 | 21.25 | 60.00 | 25.20 | 0.90 | 17.10 | 2.05 | 0.98 |
| Min | 6.00 | 1.30 | 0.60 | 18.10 | 4.20 | 0.00 | 2.00 | 0.00 | 0.00 |
| St. Dev. | 0.53 | 0.79 | 0.82 | 0.35 | 0.56 | 0.64 | 0.59 | 0.80 | 0.93 |
| Skewness | 0.14 | 0.81 | 0.86 | 0.20 | 0.81 | 0.70 | 0.71 | 0.73 | 0.64 |
| Ex. kurtosis | -1.14 | -0.74 | -0.68 | -1.29 | -0.72 | -0.58 | -0.62 | -0.87 | -0.92 |

Source: own calculations based on data from Eurostat [68].

very strong skewness in the case of $x_2$, $x_5$, $x_7$, $x_8$ and $x_9$. The distributions of seven variables $x_2$, $x_3$ and $x_5$-$x_9$ demonstrated positive kurtosis, which means a high probability of outliers. In order to eliminate the influence of outliers, a quartile criterion was applied for identification of their limits. A specified number of outliers was replaced (i.e. winsorised the tails) by a constant value. After the process of replacing values of variables the distribution of variables for winsorised data revealed weak or moderate asymmetry of all variables and small negative kurtosis. The statistical analysis suggests that in 2016 the largest percentage of the EU population struggle to pay for a one-week holiday away from home for all household members once a year ($x_1$) and with meeting unexpected expenditure ($x_4$). As regards the first variable, the highest percentage was observed in Romania (66.6%) and Croatia (61.2%) whereas the lowest was in Norway (6.0%) and (among EU members) in Sweden (8.2%). The fourth variable (being unable to meet unexpected expenditure) reached peak levels in Latvia (60.0%) and Macedonia (58.5%), with the lowest levels being recorded in Norway (18.1%) and Sweden (20.7%).

In next stage, the values of variables were normalised using the zero unitarisation transformation. After this stage all variable values are from 0 to 1. The normalised values of variables allowed us to calculate the distance of each country in European Union from the positive and negative ideal solutions. Subsequently, the values of the synthetic measure of material deprivation of countries were calculated using the TOPSIS method as the basis for identifying five types of material deprivation of countries in European Union (Tables 2 and 3, Figs 2 and 3).

Values of the synthetic measure (approach I) show the local (relative) situation of countries in the European Union. Approach I was compared with the global approach (approach II), which shows the (absolute) situation in countries of the European Union in a broader (global) context. The degree of deprivation of material needs depends on the assessment perspective–local or global. In a local approach, i.e. taking only the European context into account, countries can be divided into five groups according to the level of deprivation. In the global approach, on the other hand, there are only two groups, taking account of the global perspective. What does this mean? The explanation is based on the relativisation of the satisfaction of

**Table 2. Values of the synthetic measure and the rank of countries[a)] according to the level of material deprivation in 2016.**

| Countries | Symbol of country | Values of synthetic measures | | Rank of countries | | Level of material deprivation | |
|---|---|---|---|---|---|---|---|
| | | I | II | I | II | I local approach | II global approach |
| Bulgaria | BG | 0.930 | 0.316 | 1 | 2 | very high | low |
| Romania | RO | 0.855 | 0.295 | 2 | 3 | | |
| Serbia* | RS | 0.807 | 0.279 | 3 | 5 | | |
| Macedonia* | MK | 0.736 | 0.328 | 4 | 1 | high | |
| Greece | EL | 0.729 | 0.287 | 5 | 4 | | |
| Hungary | HU | 0.700 | 0.243 | 6 | 8 | | |
| Latvia | LV | 0.646 | 0.234 | 7 | 10 | | |
| Croatia | HR | 0.640 | 0.264 | 8 | 6 | | |
| Lithuania | LT | 0.634 | 0.238 | 9 | 9 | | |
| Portugal | PT | 0.492 | 0.201 | 10 | 11 | medium | |
| Cyprus | CY | 0.487 | 0.249 | 11 | 7 | | |
| Poland | PL | 0.459 | 0.180 | 12 | 15 | | very low |
| Slovakia | SK | 0.453 | 0.194 | 13 | 13 | | |
| Italy | IT | 0.450 | 0.200 | 14 | 12 | | |
| Ireland | IE | 0.372 | 0.187 | 15 | 14 | low | |
| Denmark | DK | 0.354 | 0.096 | 16 | 27 | | |
| Belgium | BE | 0.354 | 0.124 | 17 | 23 | | |
| Estonia | EE | 0.353 | 0.145 | 18 | 19 | | |
| Slovenia | SI | 0.332 | 0.165 | 19 | 17 | | |
| United Kingdom* | UK | 0.327 | 0.154 | 20 | 18 | | |
| Spain | ES | 0.313 | 0.179 | 21 | 16 | | |
| Malta | MT | 0.265 | 0.140 | 22 | 21 | | |
| Czechia | CZ | 0.263 | 0.142 | 23 | 20 | | |
| Finland | FI | 0.261 | 0.114 | 24 | 25 | | |
| Austria | AT | 0.246 | 0.095 | 25 | 28 | | |
| Switzerland* | CH | 0.246 | 0.081 | 26 | 31 | | |
| France | FR | 0.222 | 0.131 | 27 | 22 | | |
| Germany | DE | 0.215 | 0.117 | 28 | 24 | | |
| Norway* | NO | 0.204 | 0.066 | 29 | 33 | | |
| Iceland* | IS | 0.190 | 0.113 | 30 | 26 | very low | |
| Netherlands | NL | 0.176 | 0.093 | 31 | 29 | | |
| Luxembourg | LU | 0.090 | 0.086 | 32 | 30 | | |
| Sweden | SE | 0.051 | 0.075 | 33 | 32 | | |
| EU28 | EU | 0.437 | 0.163 | × | × | medium | very low |
| EA19 | EA19 | 0.311 | 0.153 | × | × | medium | very low |
| Max | | 0.930 | 0.328 | | | × | |
| Min | | 0.051 | 0.066 | | | | |
| Range | | 0.879 | 0.262 | | | | |
| Average | | 0.420 | 0.176 | | | | |
| Coefficient of variation (%) | | 53.99 | 42.58 | | | | |

* non-EU countries

EU28 –current composition of European Union (28 countries)–with United Kingdom; EA19 –euro area (19 countries).

[a)] Linear ordering of countries based on the values of synthetic measure obtained by: approach I–local (relative) situation, approach II–global (absolute) situation.

Source: own calculations based on data from Eurostat [68].

**Table 3. Typological classification of countries in European Union in terms of the material deprivation level in 2016.**

| Number of group | Level of material deprivation | $S_i$ | Approaches | | | |
|---|---|---|---|---|---|---|
| | | | I | | II | |
| | | | $N_g$ [a)] | % | $N_g$ | % |
| I | very high | ⟨0.80, 1.00⟩ | 3 | 9.1 | 0 | 0.0 |
| II | high | ⟨0.60, 0.80) | 6 | 18.2 | 0 | 0.0 |
| III | medium | ⟨0.40, 0.60) | 5 | 15.2 | 0 | 0.0 |
| IV | low | ⟨0.20, 0.40) | 15 | 45.5 | 12 | 36.4 |
| V | very low | ⟨0.00, 0.20) | 4 | 12.1 | 21 | 63.6 |

[a)] $N_g$ the number of $g$-th class

Source: own calculations based on data from Eurostat [68].

needs. The poorest European countries do this at a higher level than poor countries in Asia or Africa. So what for Europeans is poverty or a significant degree of deprivation can be relative prosperity for some people in the rest of the world. By comparing the two maps (Figs 2 and 3), it can be concluded that European citizens are relatively better off at meeting needs than many non-European countries. The globalisation of the problem unifies deprivation on a local scale. At the same time, it is worth noting that this does not mean that all inhabitants of European countries have high standards of living. There are still a number of groups that do not meet their minimum needs and live at a minimum level of subsistence.

Synthetic measures (approaches I and II) calculated using the classical TOPSIS method satisfactorily reflect the inter-class differences in the scope of material deprivation of countries in the European Union. Considering the ranges of variation in the case of approach I and approach II, the synthetic measure values fall within the following intervals: ⟨0.051; 0.930⟩ and ⟨0.066; 0.328⟩ respectively. All countries in the European Union in approach II and over 57.6% in approach I qualified for a class representing a very low level of material deprivation. The use of relative indicators (approach I) makes it possible to divide EU members (together with selected non-EU countries) into five groups. The use of absolute indicators (approach II) resulted in a division into only two groups (with a low or very low level of deprivation). Having in mind that deprivation mostly results from comparing one's situation to other social groups (countries), approach I was found to be better suited to presenting the differences between countries. The division based on this procedure revealed that the first group of countries (with the greatest extent of deprivation) was composed of Bulgaria, Romania and Serbia; the mean indicator value was 0.864. The second group (at a high level of material deprivation) included Macedonia, Greece, Hungary, Latvia, Croatia and Lithuania (0.681). The third group (at a medium level of deprivation) comprised Portugal, Cyprus, Poland, Slovakia and Italy (0.468). The fourth group (at a low level of deprivation) included Ireland, Denmark, Belgium, Estonia, Slovenia, the UK, Spain, Malta, the Czechia, Finland, Austria, Switzerland, France, Germany and Norway (0.288). The last group is formed by countries with the lowest level of deprivation: Iceland, the Netherlands, Sweden and Luxembourg (0.127).

Based on the analysis of the size of groups in the approach I, it may be noted that the largest one is formed by countries with a low level of deprivation, and includes over 45% of the countries covered (Table 3). From the perspective of the economic situation in Europe, it is encouraging that the groups at very high and high levels of deprivation are composed of only nine countries. Note, however, that the study did not cover certain European countries (Ukraine, Belarus, Albania or Moldova) where the standards of living are poor and which would be very likely to be included in the first group but are neither members of nor associated with the EU.

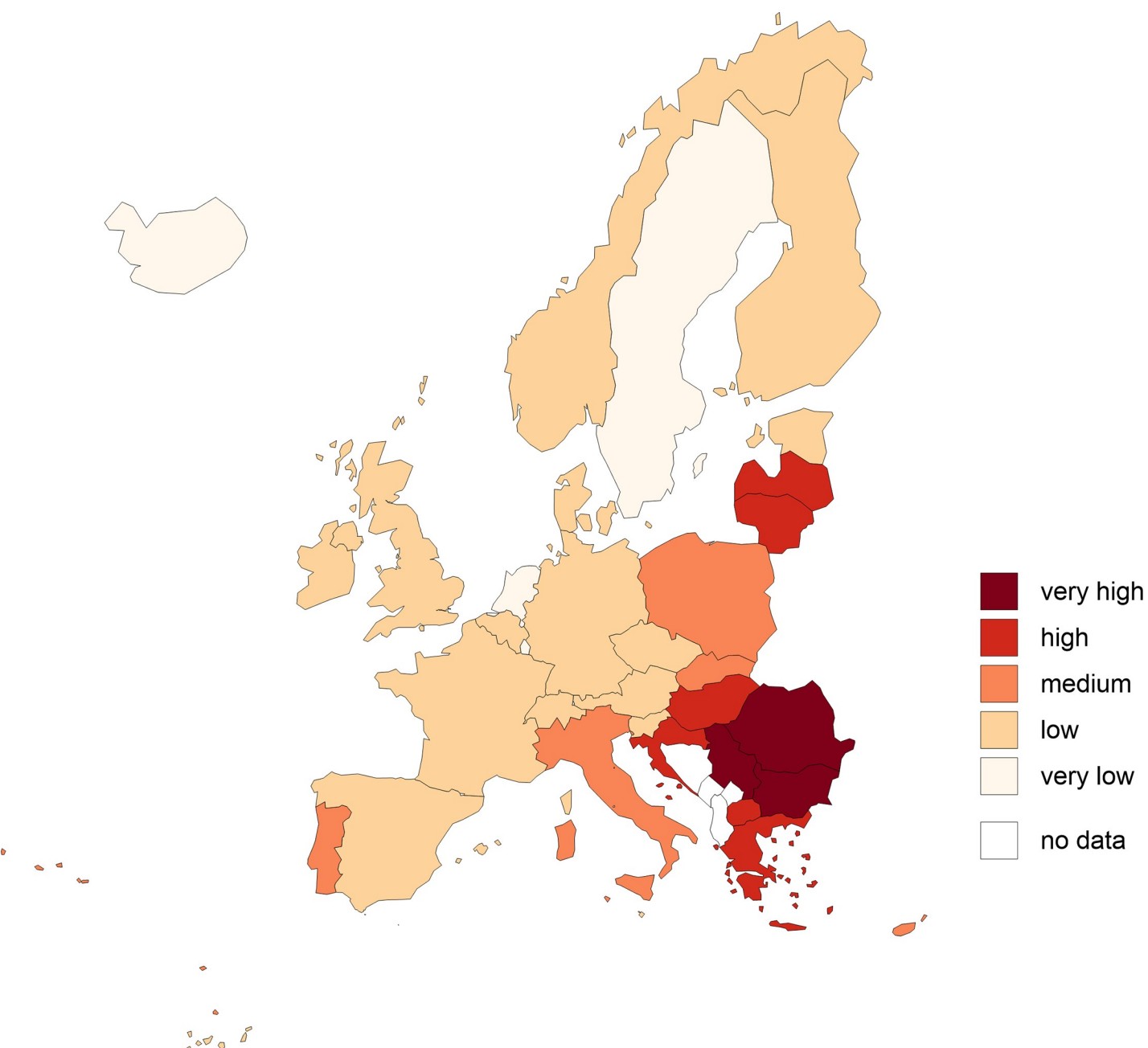

**Fig 2. Spatial delimitation of types of European Union countries according to the level of material deprivation in 2016 (local situation, approach I).** Source: own elaboration based on data from Table 2.

Table 4 presents the average values of variables for groups at various levels of deprivation. It can be noted that the greatest differences between the countries in meeting the population's material needs are when it comes to affording a one-week holiday away from home for all household members once a year. The difference in the proportion of people unable to address that need between the 1st and the 5th groups is 50.7 percentage points. Note also that owning a colour TV was the least unsatisfied need in all groups (from 0.2% in the 5th group to 1.2% in the 1st group). Irrespective of the deprivation group, it is obvious that nearly everyone in the

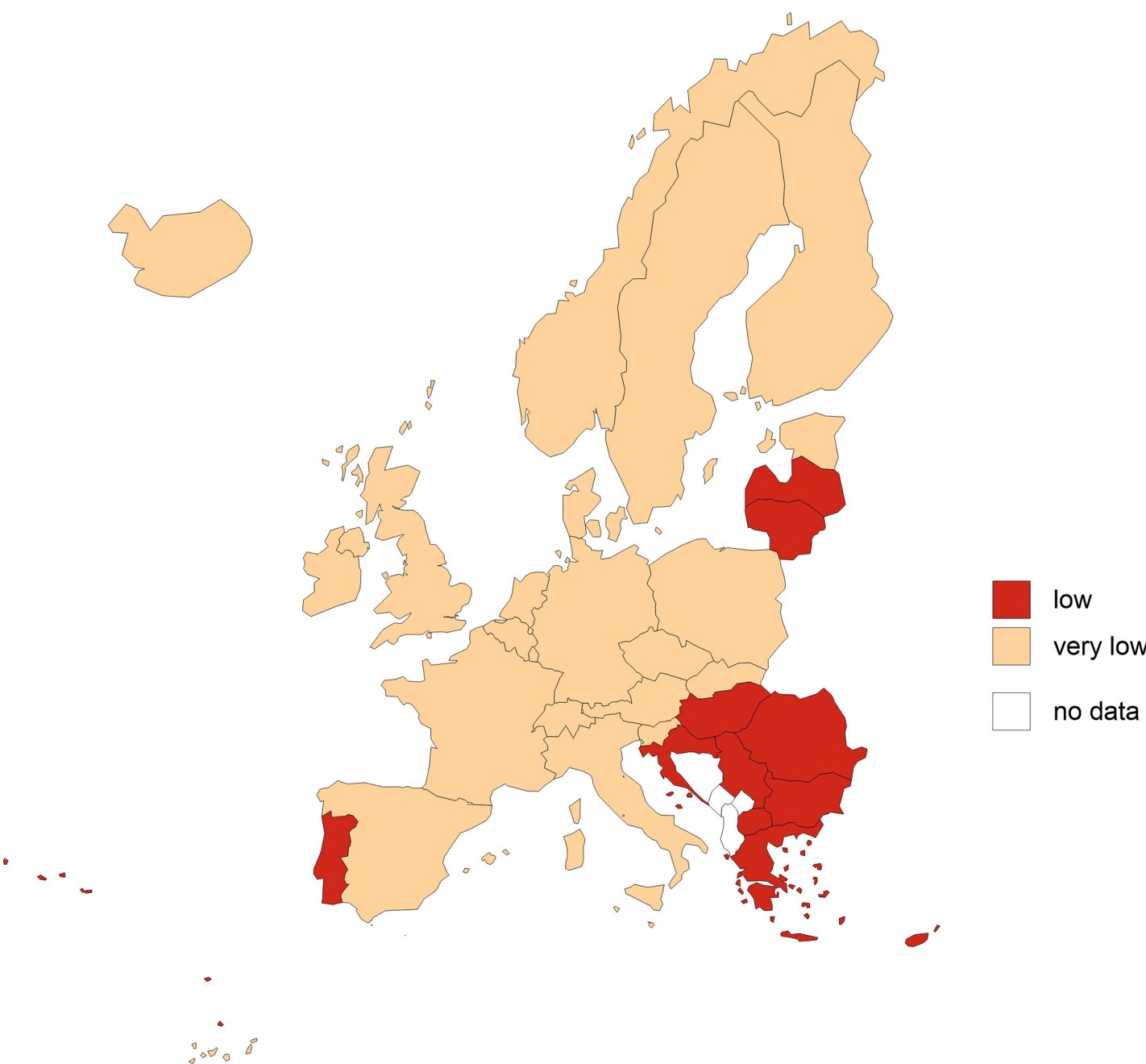

**Fig 3. Spatial delimitation of types of European Union countries according to the level of material deprivation in 2016 (global situation, approach II).** Source: own elaboration based on data from Table 2.

countries surveyed owns a TV and a phone. In recent years, these goods have become common. Meanwhile, the boundaries between common and luxury goods become blurred. This is a consequence of technological advances and higher aspirations of the population. Note also that in itself the fact that someone owns such goods as a phone, TV or washing machine does not necessarily reflect his/her living conditions. These needs are met to a relatively high degree because of consumption diffusion processes and consumption patterns being transferred to poorer classes. Note, however, that the research into deprivation only considers whether

**Table 4. Intra-class mean[a)] values of endogenous variables of material deprivation level of countries in European Union in 2016 (approach I).**

| Specification | Variable symbol | Groups of the level of material deprivation | | | | | median | EU28 | EA19 |
|---|---|---|---|---|---|---|---|---|---|
| | | I very high | II high | III medium | IV low | V very low | | | |
| 1) afford a one-week holiday away from home for all household members once a year | $x_1$ | 62.8 | 52.2 | 45.3 | 26.3 | 12.1 | 36.7 | 32.8 | 30.5 |
| 2) eat meat or fish (or a vegetarian equivalent) every second day | $x_2$ | 21.8 | 15.7 | 6.4 | 5.2 | 2.5 | 6.1 | 8.3 | 7.7 |
| 3) heat the apartment as needed | $x_3$ | 13.8 | 18.2 | 16.1 | 3.8 | 2.2 | 5.8 | 8.7 | 8.8 |
| 4) meet unexpected expenditure in the amount equal to the monthly threshold of relative poverty (as defined for the country concerned for the year preceding the survey) | $x_4$ | 54.2 | 55.7 | 38.3 | 30.0 | 22.2 | 37.9 | 36.4 | 34.6 |
| 5) pay housing-related expenses, instalments and loans on time | $x_5$ | 34.2 | 22.7 | 10.7 | 8.8 | 6.0 | 10.6 | 10.4 | 9.5 |
| 6) own a colour TV | $x_6$ | 1.2 | 0.5 | 0.3 | 0.4 | 0.2 | 0.3 | 0.4 | 0.3 |
| 7) own a car | $x_7$ | 22.0 | 15.2 | 6.9 | 6.4 | 3.2 | 6.9 | 7.7 | 5.5 |
| 8) own a washing machine | $x_8$ | 9.0 | 1.7 | 0.6 | 0.4 | 0.2 | 0.6 | 1.2 | 0.5 |
| 9) own a (fixed or mobile) phone | $x_9$ | 2.8 | 0.7 | 0.6 | 0.2 | 0.0 | 0.2 | 0.6 | 0.2 |

[a)] The mean values of variable are represented by their medians.

Source: own calculations based on data from Eurostat [68].

someone owns a certain good, disregarding its degree of wear. Although the differences in the ownership of goods alone are of minor importance, the quality and degree of wear may differ considerably between goods.

The material deprivation level refers to other variables related to the standards of living, such as overcrowding in dwellings, GDP per capita, low work intensity, poverty risk, the Gini coefficient or the level of social exclusion (Table 5). Note the interesting fact that completing one's education at a young age contributes only slightly to deprivation. Groups 2 to 4 reported similar levels of this variable; an outstanding level was recorded only in the group most severely affected by deprivation. Another important remark is that in 2016 GDP per capita in group 1 countries was nearly 60% lower than in group 5; when it comes to median equivalent income ($x_{14}$), the gap was even larger (above 90%). This suggests that there are extreme differences in the standards of living between the two groups of countries. Note also that the lower

**Table 5. Intra-class mean[a)] values of exogenous variables of socio-economic situation of countries in European Union, in 2016 (approach I).**

| Specification | Variable symbol | Typological class–the level of material deprivation | | | | | median | EU 28 | EA19 |
|---|---|---|---|---|---|---|---|---|---|
| | | I very high | II high | III medium | IV low | V very low | | | |
| Overcrowding rate | $x_{10}$ | 48.4 | 40.8 | 27.8 | 7.2 | 7.8 | 12.6 | 16.6 | 12.3 |
| Early leavers from education and training (% of population aged 18 to 24) | $x_{11}$ | 13.8 | 8.1 | 7.6 | 8.8 | 7.7 | 8.0 | 10.7 | 11.1 |
| People living in households with very low work intensity | $x_{12}$ | 11.9 | 11.6 | 9.1 | 8.4 | 7.6 | 9.1 | 10.5 | 11.1 |
| GDP aggregates per capita | $x_{13}$ | 9800.0 | 12450.0 | 16900.0 | 22600.0 | 23750.0 | 18300.0 | 20100.0 | 20900.0 |
| Median equivalised net income | $x_{14}$ | 2448.0 | 5685.5 | 8782.0 | 21713.0 | 26778.5 | 13681.0 | 16529.0 | 18240.0 |
| People at risk of poverty or social exclusion | $x_{15}$ | 38.8 | 29.3 | 25.1 | 18.4 | 17.5 | 21.9 | 23.5 | 23.1 |
| Gini coefficient of equivalised disposable income | $x_{16}$ | 37.7 | 34.0 | 32.1 | 28.6 | 27.3 | 29.5 | 30.8 | 30.7 |
| At-risk-of-poverty rate by poverty threshold | $x_{17}$ | 25.3 | 21.5 | 17.3 | 14.7 | 14.5 | 16.5 | 17.3 | 17.4 |

[a)] The mean values of variable are represented by their medians.

EU28 –current composition of the European Union (28 countries); EA19 –euro area (19 countries).

Source: own calculations based on data from Eurostat [68].

the work-intensity level, the more likely it is for a person to become a member of the population least able to meet their needs.

## Discussion

It is also worth considering how state policies can affect deprivation, especially in the context of unequal participation of the population in GDP, or counteracting income stratification (measured by the Gini index). Research indicates [69] that the main areas of such policies (mezo and macro dimensions) are 1) market institutions, 2) civil rights, 3) civic values. In the sphere of market institutions, it is necessary to indicate those which generate opportunities to improve the income situation and, consequently, reduce material deprivation. These institutions should primarily include the labour market, together with income policy, and support institutions. The activity of institutions should be considered in the context of their impact on macroeconomic variables: GDP, unemployment rate, but also income redistribution etc. The scope of activity of state institutions results from civil rights. These include mechanisms that counteract social and economic exclusion and unacceptable levels of stratification. In this context, we should also ask ourselves the consequences of the rights-based approach (to social transfers, to benefits, etc.) in terms of reducing deprivation, namely, do they actually prevent it? Or do they perpetuate poverty by making benefit recipients dependent on aid instead of activating them [70]? In turn, within the framework of civic values, one should take account of inclusive actions resulting from the formation of civil society.

The interaction between these three areas differs in different European countries, and there are many manifestations of policies limiting material deprivation (e.g. recent Covid-related actions of individual governments to counteract the deepening of poverty). However, it is difficult to talk about universal solutions that could work in all countries. Policies should be conducted at the local level, then their impact is widest.

Nevertheless, the comparability of European statistics requires similar actions in all countries.The question therefore arises of whether the characteristics of material deprivation should not be developed or changed in the coming years. If so, to what extent? It seems that some of the goods should be changed. Goods such as a colour television, an automatic washing machine or a telephone have become commonplace. Even poor people who are socially marginalised are very often in possession of these goods. On the other hand, there are groups of people who consciously dispense with these goods–"zero waste" groups, or pro-ecological groups. This approach can wrongly include those who are not really poor among the excluded. Perhaps it would be worth considering whether these goods should not be replaced by others in the study of material deprivation–a dishwasher, access to the Internet (preventing digital exclusion)–or perhaps access to services, such as hairdressing, financial services, public transport or the ability to a meal outside the home several times a month, which are still treated as elements of wealth in many countries. It is also worth considering whether a factor of social exclusion is inadequate provision of technical infrastructure or limited access to sewage and water supply. The question of selecting goods and services remains open.

There is a need for social politicians and researchers of poverty to discuss which of these are more decisive for present-day poverty. It is worth stressing that contemporary poverty is changing, so indicators should also be constantly changing. And although this will make comparability over time more difficult, it will more accurately reflect the actual sphere of deprivation. It is obvious that the more extensive the indicator, the more difficult it is to interpret, while at the same time it becomes less transparent. It is difficult to return entirely to the concept of relative deprivation by Peter Townsend [71] or John Veit-Vilson [16], which, however correct and able to determine the level of deprivation, were nevertheless too extensive.

Townsend proposed sixty indicators of standards of living and lifestyles within twelve groups of needs, such as food, clothing, fuel and light, home furnishings, dwellings and facilities, immediate home environment, general working conditions and safety at work, family support, recreation, education, health and social relations. He proposed the social verification of the list of unmet needs, which was initially prepared by experts. Veit-Wilson pointed out the need to value social assessments. It seems that nowadays we should focus on the subjective aspects of material deprivation, because poverty is subjective. It varies depending on the context, place or the possibilities to deal with it.

## Conclusion

The modified positional TOPSIS was used to synthetically assess and identify the types of material deprivation levels in different countries. This is a suitable approach to determine the synthetic measure in a case where the set of variables for EU countries includes strong asymmetry and atypical values (outliers). The division of countries surveyed into types by deprivation level showed that there are five different typological groups. They primarily differ in the extent to which the nine basic needs (reflecting the level of material deprivation) are addressed. The first group, at the highest level of deprivation, includes Bulgaria and Romania, i.e. countries who joined the EU as a result of the last but one enlargement. They reported the lowest standard-of-living indicators. The first group also included Serbia, currently a candidate member. The second group (at a high level of material deprivation) was composed of five EU countries and another candidate member, Northern Macedonia. Among EU members, this group included Greece, which is struggling with a serious economic crisis. Other members in this class are Hungary, Latvia and Lithuania, who joined the EU in 2004, and Croatia, an EU member since 2013. The third group (at a medium level of deprivation) included Portugal, Cyprus, Poland, Slovakia and Italy. The fourth group (at a low level of deprivation) consisted of fifteen countries, mostly old EU members (Ireland, Denmark, Belgium, Estonia, Slovenia, the UK, Spain, Malta, Czechia, Finland, Austria, Switzerland, France, Germany and Norway) which demonstrate high standards of living and enjoy a relatively good economic standing. The last group was formed by countries at the lowest level of deprivation: Iceland, the Netherlands, Sweden and Luxembourg. These are generally considered to offer high standards of living.

As the material deprivation indicator itself is of a declarative nature, it is strongly impacted by reference groups. Based on the Easterlin concept [72], it may be assumed that residents of wealthy countries take their economic condition for granted and do not consider it to be particularly good. Usually, they compare their economic well-being with people in a more advantageous position, and the goals they pursue generally exceed their current capabilities. Hence they may be subjectively dissatisfied with their living conditions. A favourable economic standing may drive aspirations, and therefore the interviewees may feel their situation leaves something to be desired. At the same time, the opposite situation may occur: the respondents in poorer countries may be relatively less likely to feel their needs are not sufficiently addressed.

Note also that the division of countries into deprivation classes is correlated with characteristics that strongly impact the perceived standards of living: overcrowding of dwellings, GDP per capita, low work intensity, poverty risk, the Gini coefficient or the level of social exclusion.

## Recommendations

The proposed research approach may be the basis for the establishment of programme documents in the EU, e.g. development strategies or programmes for poverty reduction and

lowering material deprivation. This study may provide the stimulus for new directives to be adopted by institutions in charge of social policy in EU countries.

Note that, although somewhat utopian, these provisions play an important role in improving labour and social conditions and raising minimum wages. Otherwise, the polarisation between new and old EU members (or between the EU and its close collaborators) will continue to be noticeable.

## Supporting information

**S1 Data.**
(XLS)

## Author Contributions

**Conceptualization:** Aleksandra Łuczak, Sławomir Kalinowski.

**Data curation:** Aleksandra Łuczak, Sławomir Kalinowski.

**Formal analysis:** Aleksandra Łuczak, Sławomir Kalinowski.

**Funding acquisition:** Aleksandra Łuczak.

**Methodology:** Aleksandra Łuczak, Sławomir Kalinowski.

**Resources:** Aleksandra Łuczak, Sławomir Kalinowski.

**Supervision:** Sławomir Kalinowski.

**Writing – original draft:** Sławomir Kalinowski.

**Writing – review & editing:** Aleksandra Łuczak, Sławomir Kalinowski.

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
