## [Decision Letter · Decision Letter 0]

27 May 2020

PONE-D-20-01799

Assessing the level of the material deprivation of European Union countries

PLOS ONE

Dear Dr. Kalinowski,

Thank you for submitting your manuscript to PLOS ONE. After careful consideration, we feel that it has merit but does not fully meet PLOS ONE’s publication criteria as it currently stands. Therefore, we invite you to submit a revised version of the manuscript that addresses the points raised during the review process.

We look forward to receiving your revised manuscript.

Kind regards,

Fausto Cavallaro, PhD

Academic Editor

PLOS ONE

Journal Requirements

3. Thank you for stating the following financial disclosure: 'NO'

Reviewers' comments:

Reviewer's Responses to Questions

**Comments to the Author**

1. Is the manuscript technically sound, and do the data support the conclusions?

Reviewer #1: Yes

Reviewer #2: Yes

Reviewer #3: Yes

2. Has the statistical analysis been performed appropriately and rigorously? 

Reviewer #1: Yes

Reviewer #2: Yes

Reviewer #3: Yes

3. Have the authors made all data underlying the findings in their manuscript fully available?

Reviewer #1: Yes

Reviewer #2: Yes

Reviewer #3: No

4. Is the manuscript presented in an intelligible fashion and written in standard English?

Reviewer #1: Yes

Reviewer #2: Yes

Reviewer #3: Yes

5. Review Comments to the Author

Reviewer #1: The article is well-written and deals with important issues related to poverty, understood here as the lack of sufficient material resources to meet the needs of the individual.

The topic is interesting, not often raised. The material dimension of poverty is more difficult to operationalize than the income dimension, hence it is less frequently studied. Therefore, the text deserves to be published.

Work structure appropriate.

Some comments:

1. The chapter 'Literature Review' should be supplemented with a footnote on the basis of which research or EU document it analyzes poverty with these three measures. - page 4.

2. In addition, I want to emphasize that the authors first of all assess the relative material deprivation in local terms as the situation of countries in the European Union. Secondly - which is an asset - it shows the (absolute) situation in the European Union in a broader (global) context.

And this aspect should be complemented in the aim of the study presented in the INRODUCTION. I propose to write: The purpose of the work is to assess the level of material deprivation in the European Union in 2016 in a local perspective and in a global perspective. The purpose of this paper is to assess the level of material deprivation in European Union countries in 2016 with global approach (approach II), which shows (absolute) situation in countries of the European Union in a broader (global) context.

3. On page 5 the authors write: An interesting piece of research information can be

derived from the relationship between material deprivation and the at-risk-of-poverty rate represented by 60% of the national median income.

This is very interesting, and it is worth supplementing the discussion with this dependence. The more so as the authors use this indicator in Table 5. In addition it is worth adding after the aim of the study (as above) that the effort was taken also to seek the relation between the assessment of the level of material deprivation in EU countries with the at-risk-of-poverty rate.

4. I propose that the results in Table 2 should be presented graphically in a cartogram. For readers outside Europe, interpretation of the results will then be easier. And the visualization of test results will enrich the text.

5. In table 2, in the column "Level of material deprivation” the authors should add: I- local aproach, II - global approach.

Reviewer #2: This is a useful analysis, well presented and documented. I particularly commend the authors on the clarity of explanation of the analytical procedures (such as normalization, etc.). The conclusions are clear.

However the following points deserve discussion:

(a) Is there a relationship between GDP (or per-capita GDP) and the indices? If yes or no or weak, what are the implications?

(b) How can policy/program variables influence deprivation directly or indirectly? Can the authors suggest past or future experiments (such as inclusion of states in the EU, or the effects of Covid19 slowing) that might provide useful clues?

Reviewer #3: The paper “Assessing the level of the material deprivation of European Union countries” addresses a challenging issue (how to measure individual well-being in Europe through one of its dimensions, i.e. material deprivation) with important policy implications.

I have two major reservations:

The first refers to the empirical methodology. The method used in the paper (i.e. TOPSIS) should be explained clearly for a general reader. I would not assume everybody is familiar with this technique. The same applies to other concepts presented in the Methods section (e.g. stimulants, destimulants, nominants, …).

At stage two it seems that the objects of the analysis are defined as “countries”? Is it so? How is the heterogeneity of individuals provided by microlevel data exploited?

It is not clear which release of microlevel data (i.e. EUSILC) is used.

The second reservation relates to the added value of this empirical methodology implemented for the first time (to my knowledge) to this domain: what does it add to the relevant literature? Which are the policy implications of the results derived in this paper with respect to those derived in the widely cited literature?

Minor points:

Abstract: avoid the use of acronyms (i.e. TOPSIS) if not explained yet

Page 9: 33 countries are mentioned as European countries: it would nice to mention the countries included in the analysis in addition to the EU member states (presented in Table 2)

Table 2: there is a mismatch between country name and country acronym (e.g. Italy LV): please check numbers as well

Conclusion: references to “municipalities” is not clear.

6. PLOS authors have the option to publish the peer review history of their article (what does this mean?). If published, this will include your full peer review and any attached files.

Reviewer #1: Yes: Monika Stanny

Reviewer #2: Yes: Shripad Tuljapurkar

Reviewer #3: No

---

## [Author Response · Author response to Decision Letter 0]

24 Jul 2020

Dear Editors,

Subject: Assessing the level of the material deprivation of European Union countries

Thank you for your e-mail dated 27 May 2020 enclosing reviews. Thank you also Reviewers for their constructive comments and helpful suggestion on the manuscript. We have carefully reviewed the comments and have revised the manuscript accordingly. Our responses are given below in the tables. Below there are the Reviewer's comments (in grey) and our responses with description on how the manuscript was revised (in black).

We hope the revised version is now suitable for publication in PLOS ONE.

Best regards,

Aleksandra Łuczak

Sławomir Kalinowski

Authors' Responses to Reviews 

First of all, we would like to thank Reviewers for the comments. It was a great help in improving our paper. All comments proposed by the reviewers have been written on the text (in grey). At the same time, we described below what we did.

Reviewers' comments:

1. Is the manuscript technically sound, and do the data support the conclusions?

Reviewer #1: Yes

Reviewer #2: Yes

Reviewer #3: Yes

2. Has the statistical analysis been performed appropriately and rigorously?

Reviewer #1: Yes

Reviewer #2: Yes

Reviewer #3: Yes

3. Have the authors made all data underlying the findings in their manuscript fully available?

Reviewer #1: Yes

Reviewer #2: Yes

Reviewer #3: No

Response: We did our best to present all data underlying the findings in our manuscript fully available. We present them briefly and clearly. Data comes from Eurostat. In the Results of research we also present descriptive statistics and intra-class mean values of variables of material deprivation level and socio-economic situation. 

All data used for the calculation shall be taken from published Eurostat data, including in particular EU-SILC. Tables 1 and 4 use ilc_mdes, while Table 5 uses ilc_mddd, ilc_mdsd, ilc_li, ilc_iei.

https://ec.europa.eu/eurostat/web/income-and-living-conditions/data/database

4. Is the manuscript presented in an intelligible fashion and written in standard English?

Reviewer #1: Yes

Reviewer #2: Yes

Reviewer #3: Yes

 

Responses to Reviewer #1 Comments

The article is well-written and deals with important issues related to poverty, understood here as the lack of sufficient material resources to meet the needs of the individual.

The topic is interesting, not often raised. The material dimension of poverty is more difficult to operationalize than the income dimension, hence it is less frequently studied. Therefore, the text deserves to be published.

Work structure appropriate.

Dear Reviewer

Thank you very much for revising our manuscript entitled “Assessing the level of the material deprivation of European Union countries”. 

Your suggestions are valuable and significant in improving the overall quality of our paper. We analyzed the comments carefully and corrected the manuscript accordingly. In the text below we would like to explain how we revised the manuscript based on your suggestions and try to dispel your doubts.

Some comments:

Point 1. The chapter 'Literature Review' should be supplemented with a footnote on the basis of which research or EU document it analyzes poverty with these three measures. - page 4.

Response: Thank you for your comments, we completed the document - it was about the document, of course: 

COMMUNICATION FROM THE COMMISSION, EUROPE 2020. A strategy for smart, sustainable and inclusive growth, Brussels, 3.3.2010 COM(2010) 2020.

Point 2. In addition, I want to emphasize that the authors first of all assess the relative material deprivation in local terms as the situation of countries in the European Union. Secondly - which is an asset - it shows the (absolute) situation in the European Union in a broader (global) context.

And this aspect should be complemented in the aim of the study presented in the INRODUCTION. I propose to write: The purpose of the work is to assess the level of material deprivation in the European Union in 2016 in a local perspective and in a global perspective. The purpose of this paper is to assess the level of material deprivation in European Union countries in 2016 with global approach (approach II), which shows (absolute) situation in countries of the European Union in a broader (global) context.

Response: We corrected the purpose in the Introduction and also in the Abstract as suggested by the Reviewer. We added in the purpose in a local perspective and in a global perspective.

“The purpose of this paper is to assess the level of material deprivation in European Union countries in 2016 from both a local and global perspective. An effort was also made to seek the relationship between the assessment of the level of material deprivation in EU countries and the at-risk-of-poverty rate. Identifying the major poverty factors could provide a basis for designing tools that reduce the risk of poverty and social exclusion in various EU countries. In this context, note that the social integration process in the EU is based on the open method of coordination (OMC), which provides member states with considerable discretion in choosing the measures and priorities when carrying out their tasks. Hence it is important to identify the levels of material deprivation in different countries of the European Union as one of the methods for diagnosing their social situation.

The authors’ contribution includes, firstly, a proposal for quantitative approaches based on the Technique for Order Preference by Similarity to an Ideal Solution (TOPSIS) to assess the material deprivation level in EU countries. Secondly, the authors have improved the quality of assessment of the material deprivation level in EU countries and expanded possibilities of research from two perspectives (local and global). The local approach presents the relative situation of countries in the EU. The global approach shows the absolute situation of EU countries, i.e. in a broader (global) context. The results of the study are important for decision-makers and politicians in EU who participate in the process of creating documents (polices, financial plans, strategies).”

Point 3. On page 5 the authors write: An interesting piece of research information can be derived from the relationship between material deprivation and the at-risk-of-poverty rate represented by 60% of the national median income.

This is very interesting, and it is worth supplementing the discussion with this dependence. The more so as the authors use this indicator in Table 5. In addition it is worth adding after the aim of the study (as above) that the effort was taken also to seek the relation between the assessment of the level of material deprivation in EU countries with the at-risk-of-poverty rate.

Response: We also added in the Introduction as suggested by the reviewer: The effort was taken also to seek the relation between the assessment of the level of material deprivation in EU countries with the at-risk-of-poverty rate. 

We were going to present in the text the relationship between material deprivation and the at-risk-of-poverty rate, however, the results were not sufficiently convincing to us.

After initial attempts with model construction we have decided that in each of the approaches the impact is so low that we will not present these relationships in the article. In addition, the set of external features is so extensive that creating a model (e.g. ordered logit model) including all variables is the problem for new broader research. At the same time, as suggested by the EU, it should be easy and simple to calculate.

Point 4. I propose that the results in Table 2 should be presented graphically in a cartogram. For readers outside Europe, interpretation of the results will then be easier. And the visualization of test results will enrich the text.

Response: We have also added two cartograms on spatial delimitation of types of countries according to the level of material deprivation in 2016. Cartograms preset local situation (approach I) and global situation (approach II) of EU countries.

Thank you for this suggestion, in fact the cartogram shows to a greater extent where the problem of material deprivation exists in Europe. We illustrated this with both the cartogram and the text.

We have presented two cartograms - local and global situation, which will make it clear to readers that deprivation of needs is relative. If we assess it from a European perspective, there are differences between the prosperous western part of the EU and central and eastern Europe, where material deprivation is greater. Showing these two cartograms makes us realise that countries that are poor in Europe and do not realise needs at the average level for the EU can at the same time be considered globally prosperous.

Figure 1. Spatial delimitation of types of European Union countries according to the level of material deprivation in 2016 (local situation, approach I).

Source: own elaboration based on data from table 2.

 

Figure 2. Spatial delimitation of types of European Union countries according to the level of material deprivation in 2016 (global situation, approach II).

Source: own elaboration based on data from table 2.

Point 5. In table 2, in the column "Level of material deprivation” the authors should add: I- local aproach, II - global approach.

Response: Thank you for comment. We added in table 2, in the column "Level of material deprivation”: I - local approach, II - global approach.

We also added a text explaining the global context and local deprivation of needs.

The degree of deprivation of material needs depends on the assessment perspective – local or global. In a local approach, i.e. taking only the European context into account, countries can be divided into five groups according to the level of deprivation. In the global approach, on the other hand, there are only two groups, taking account of the global perspective. What does this mean? The explanation is based on the relativisation of the satisfaction of needs. The poorest European countries do this at a higher level than poor countries in Asia or Africa. So, what for Europeans is poverty or a significant degree of deprivation can be relative prosperity for some people in the rest of the world. By comparing the two maps (Figures 1 and 2), it can be concluded that European citizens are relatively better off at meeting needs than many non-European countries. The globalisation of the problem unifies deprivation on a local scale. At the same time, it is worth noting that this does not mean that all the inhabitants of European countries have high standards of living. There are still a number of groups that do not meet their minimum needs and live at a minimum level of subsistence.  

Responses to Reviewer #2 Comments

Point 1. This is a useful analysis, well presented and documented. I particularly commend the authors on the clarity of explanation of the analytical procedures (such as normalization, etc.). The conclusions are clear.

Response: Thank you so much for such a kind words of appreciation. Every feedback is valuable to us and brings us closer to a satisfactory level.

Point 2. However the following points deserve discussion:

(a) Is there a relationship between GDP (or per-capita GDP) and the indices? If yes or no or weak, what are the implications?

Response: For the purposes of this article, we tried to create a logit model. The identified classes of material deprivation levels of the EU countries became the starting point for the econometric analysis. They represented ordered categories of the explained variable in a logit model combined with a system of variables of socio-economic situation (including GDP). The increasing value of explanatory variables (assuming ceteris paribus), such as the At-risk-of-poverty rate by poverty threshold and the Overcrowding rate, increased the chance of changing the level of material deprivation for the worse (by 41.0% and 11.3% respectively). GDP aggregates per capita, on the other hand, had a stimulating effect (<1%) on reducing the level of material deprivation in EU countries. We therefore decided that there was no point in inserting such an estimated model into the article.

Point 3. (b) How can policy/program variables influence deprivation directly or indirectly? Can the authors suggest past or future experiments (such as inclusion of states in the EU, or the effects of Covid19 slowing) that might provide useful clues?

Response: In the "discussion" section, we added reflections about policy and its influence on material deprivation and poverty. We also added considerations on the material deprivation indicator.

It is also worth considering how state policies can affect deprivation, especially in the context of unequal participation of the population in GDP, or counteracting income stratification (measured by the Gini index). Research indicates [69] that the main areas of such policies (mezo and macro dimensions) are 1) market institutions, 2) civil rights, 3) civic values. In the sphere of market institutions, it is necessary to indicate those which generate opportunities to improve the income situation and, consequently, reduce material deprivation. These institutions should primarily include the labour market, together with income policy, and support institutions. The activity of institutions should be considered in the context of their impact on macroeconomic variables: GDP, unemployment rate, but also income redistribution, etc. The scope of activity of state institutions results from civil rights. These include mechanisms that counteract social and economic exclusion and unacceptable levels of stratification. In this context, we should also ask ourselves the consequences of the rights-based approach (to social transfers, to benefits, etc.) in terms of reducing deprivation, namely, do they actually prevent it? Or do they perpetuate poverty by making benefit recipients dependent on aid instead of activating them [70]? In turn, within the framework of civic values, one should take account of inclusive actions resulting from the formation of civil society.

The interaction between these three areas differs in different European countries, and there are many manifestations of policies limiting material deprivation (e.g. recent Covid-related actions of individual governments to counteract the deepening of poverty). However, it is difficult to talk about universal solutions that could work in all countries. Policies should be conducted at the local level, then their impact is widest.

Nevertheless, the comparability of European statistics requires similar actions in all countries. The question therefore arises of whether the characteristics of material deprivation should not be developed or changed in the coming years. If so, to what extent? It seems that some of the goods should be exchanged. Goods such as a colour television, an automatic washing machine or a telephone have become commonplace. Even poor people who are socially marginaliszed are very often in possession of these goods. On the other hand, there are groups of people who consciously dispense with these goods –- “"zero waste”" groups, or pro-ecological groups. This approach can wrongly include those who are not really poor among the excluded. Perhaps it would be worth considering whether these goods should not be replaced by others in the study of material deprivation -– a dishwasher, access to the Internet (preventing digital exclusion) –, or perhaps access to services, such as - hairdressing, financial services, public transport or the ability to a meal outside the home several times a month, which are still treated as elements of wealth in many countries. It is also worth considering whether a factor of social exclusion is inadequate provision of technical infrastructure or limited access to sewage and water supply. The question of selecting goods and services remains open. 

There is a need for social politicians and researchers of poverty to discuss, which of these are more decisive for present-day poverty. It is worth stressing that contemporary poverty is changing, so indicators should also be constantly changing. And although this will make comparability over time more difficult, it will more accurately reflect the actual sphere of deprivation. It is obvious that the more extensive the indicator, the more difficult it is to interpret, while at the same time it becomes less transparent. It is difficult to return entirely to the concept of relative deprivation by Peter Townsend [71] or John Veit-Vilson [16], which, however correct and able to determine the level of deprivation, were nevertheless too extensive. Townsend proposed sixty indicators of standards of living and lifestyles within twelve groups of needs, such as: food, clothing, fuel and light, home furnishings, dwellings and facilities, immediate home environment, general working conditions and safety at work, family support, recreation, education, health and, social relations. He proposed a the social verification of the list of unmet needs, which was initially prepared by experts. Veit-Wilson pointed out the need to value social assessments. It seems that nowadays we should focus on the subjective aspects of material deprivation, because poverty is subjective. It varies depending on the context, place or the possibilities to deal with it. 

Responses to Reviewer #3 Comments

Point 1. The paper “Assessing the level of the material deprivation of European Union countries” addresses a challenging issue (how to measure individual well-being in Europe through one of its dimensions, i.e. material deprivation) with important policy implications.

I have two major reservations:

The first refers to the empirical methodology. The method used in the paper (i.e. TOPSIS) should be explained clearly for a general reader. I would not assume everybody is familiar with this technique. The same applies to other concepts presented in the Methods section (e.g. stimulants, destimulants, nominants, …).

Response: We did our best to present the TOPSIS method clearly. We have expanded the following description in the Methods:

The positional TOPSIS (Technique for Order Preference by Similarity to an Ideal Solution) approach was proposed in order to assess the level of material deprivation in European Union countries (selected non-EU but closely related countries were also taken into account). The classical version of the TOPSIS method was developed by Hwang and Yoon [57] and is one of the best-known techniques for solving multi-criteria decision-making problems with a finite number of alternatives. TOPSIS is very useful in constructing the ranking of objects described by many variables. It is based on the distances of objects from the ideal solution and the anti-ideal solution. Distances are the basis for constructing a synthetic measure. In the assessment of the material deprivation of EU countries, variables characterised by strong asymmetry and atypical values (outliers) may occur. In such cases the classical TOPSIS method may be unreliable and contribute to problems connected with the complete and accurate identification of types of material deprivation. In studies concerning material deprivation the focus should thus be on robust methods that limit the impact of strong asymmetry and outliers, particularly those using the Weber spatial median [58]. It should be added that TOPSIS and its modification and various versions have been widely used in many issues [59-61] i.e. business, management, health, safety, environment and many others. We also explained terms i.e. stimulants, destimulants, nominants as follows:

The selected variables are classified as stimulants, destimulants and nominants. They have a stimulating or destimulating effect on the phenomenon. Variables that have stimulating effect contribute to increasing the level of the phenomenon and are called stimulants, while variables with a destimulating effect decrease it and are called destimulants. Nominants are the type of variables that are destimulants in one range of a variable and stimulants in another. Desirable (optimal) values should be defined for the nominants. Point 2. At stage two it seems that the objects of the analysis are defined as “countries”? Is it so? How is the heterogeneity of individuals provided by microlevel data exploited?

Response: We would like to thank Reviewer for valuable comment, which encourages reflection on expand our analyses and can be an inspiration for further research. We would also like to clarify, that in this study, we used aggregated data for EU countries, not microlevel data. For this reason, the research of heterogeneity of individuals could not be the subject of our research.

Point 3. It is not clear which release of microlevel data (i.e. EUSILC) is used.

Resnponse: All data used for the calculation shall be taken from published Eurostat data, including in particular EU-SILC. Tables 1 and 4 use ilc_mdes, while Table 5 uses ilc_mddd, ilc_mdsd, ilc_li, ilc_iei.

These data are available at https://ec.europa.eu/eurostat/web/income-and-living-conditions/data/database

Point 4. The second reservation relates to the added value of this empirical methodology implemented for the first time (to my knowledge) to this domain: what does it add to the relevant literature? Which are the policy implications of the results derived in this paper with respect to those derived in the widely cited literature?

Response: We also added in the Introduction following sentences:

“The authors’ contribution includes, firstly, a proposal for quantitative approaches based on the Technique for Order Preference by Similarity to an Ideal Solution (TOPSIS) to assess the material deprivation level in EU countries. Secondly, the authors have improved the quality of assessment of the material deprivation level in EU countries and expanded possibilities of research from two perspectives (local and global). The local approach presents the relative situation of countries in the EU. The global approach shows the absolute situation of EU countries, i.e. in a broader (global) context. The results of the study are important for decision-makers and politicians in EU who participate in the process of creating documents (polices, financial plans, strategies).”.

Discussions were complemented by reflections on the material deprivation rate itself and the need to change it. 

The policy implications are presented in the first paragraph of the "discussions".

Minor points:

Abstract: avoid the use of acronyms (i.e. TOPSIS) if not explained yet

Thank you for notice. In abstract we explained acronym TOPSIS.

Page 9: 33 countries are mentioned as European countries: it would nice to mention the countries included in the analysis in addition to the EU member states (presented in Table 2)

Table 2: there is a mismatch between country name and country acronym (e.g. Italy LV): please check numbers as well

Conclusion: references to “municipalities” is not clear.

We exchanged “municipalities” for “EU countries”. This mistake was due to an incorrect translation.

The mistake concerning the abbreviations of countries' names has been removed. It was due to an oversight.

---

## [Decision Letter · Decision Letter 1]

17 Aug 2020

Assessing the level of the material deprivation of European Union countries

PONE-D-20-01799R1

Dear Dr. Kalinowski,

We’re pleased to inform you that your manuscript has been judged scientifically suitable for publication and will be formally accepted for publication once it meets all outstanding technical requirements.

Kind regards,

Fausto Cavallaro, PhD

Academic Editor

PLOS ONE

Reviewers' comments:

Reviewer's Responses to Questions

**Comments to the Author**

1. If the authors have adequately addressed your comments raised in a previous round of review and you feel that this manuscript is now acceptable for publication, you may indicate that here to bypass the “Comments to the Author” section, enter your conflict of interest statement in the “Confidential to Editor” section, and submit your "Accept" recommendation.

Reviewer #1: All comments have been addressed

2. Is the manuscript technically sound, and do the data support the conclusions?

Reviewer #1: Yes

3. Has the statistical analysis been performed appropriately and rigorously? 

Reviewer #1: Yes

4. Have the authors made all data underlying the findings in their manuscript fully available?

Reviewer #1: Yes

5. Is the manuscript presented in an intelligible fashion and written in standard English?

Reviewer #1: Yes

6. Review Comments to the Author

Reviewer #1: All comments that I made have been taken into account in the text. I recommend the text for publication.

7. PLOS authors have the option to publish the peer review history of their article (what does this mean?). If published, this will include your full peer review and any attached files.

Reviewer #1: **Yes: **Monika Stanny

---

## [Editor Report · Acceptance letter]

21 Aug 2020

PONE-D-20-01799R1 

Assessing the level of the material deprivation of European Union countries 

Dear Dr. Kalinowski:

I'm pleased to inform you that your manuscript has been deemed suitable for publication in PLOS ONE. Congratulations! Your manuscript is now with our production department. 

Kind regards, 

on behalf of

Professor Fausto Cavallaro 

Academic Editor

PLOS ONE